# Women's experiences with the use of traditional medicine during childbirth in selected areas of Zambia

Wanga Zulu[1,2]*, Joseph Mumba Zulu[3], Charles Michelo[2], Choolwe Jacobs[2]

1 Community Health Unit, Department of Public Health, Ministry of Health, Lusaka, Zambia, 2 Department of Epidemiology and Biostatistics, School of Public Health, University of Zambia, Lusaka, Zambia, 3 Department of Health Policy and Management, School of Public Health, University of Zambia, Lusaka, Zambia

* zulunkuwa@gmail.com

## Abstract

While motherhood is often a positive and fulfilling experience, for many women, it is associated with suffering, pain, ill-health, and even death. Like in many African countries, some Zambian communities rely on traditional medicines to mitigate birth related challenges and make motherhood a more positive experience. Most researchers emphasise that the safety of traditional medicines is particularly important for pregnant women. This qualitative study was conducted to better understand women's experiences and perceptions of traditional medicine in maternal care, particularly how they navigate its use alongside modern medical practices in selected areas of Zambia. The study explored women's perceptions and experiences towards the use of traditional medicines during childbirth in Zambia. This was an explanatory study, with sixteen focused group discussions conducted with women that used traditional medicine during childbirth (n = 192) and eight with women who supply traditional medicine to pregnant women (n = 96). The sample was purposively recruited until data saturation was reached. The study was conducted in Lusaka, Solwezi and Kaoma districts of Zambia. Data were coded and organised using NVivo 10 (QSR international), and were analysed using thematic analysis. While some women reported faster labour progression and less pain, others experienced complications. Cultural beliefs, fear of caesarean births, and hospital mistreatment reinforced the reliance on traditional medicine. Traditional medicine was commonly administered through oral ingestion, vaginal insertion, and medicinal tattooing to, restore fertility, for vaginal tightness, prevent complications and aid postpartum recovery. Many women perceive herbal remedies safer alternatives to conventional medical treatments, especially in settings where access to formal healthcare was a challenge. Women's experiences with herbal medicine during

**Data availability statement:** Due to ethical and privacy concerns, the full qualitative data set (transcripts from focus group discussions) cannot be made publicly available. However, dede-identified excerpts relevant to the study findings are available upon reasonable request. Interested researchers may contact the corresponding author, Wanga Zulu (zulunkuwa@gmail.com), to request access to the data. In an event that the Corresponding Author is unavailable kindly contact Sody Mweetwa Munsaka, BSc., MSc., PhD Chairperson University of Zambia Biomedical Research Ethics Committee Tel: +260977925304 Email s.munsaka@unza.zm Ridgeway Campus P.O. Box 50110 University of Zambia Lusaka, Zambia.

**Funding:** The authors received no specific funding for this work.

**Competing interests:** The authors have declared that no competing interests exist.

childbirth are shaped by a combination of personal beliefs, cultural traditions, and the socio-economic context in which they live. These findings highlight the need for safe and culturally sensitive maternal healthcare interventions.

## Introduction

Women's use of herbal medicine during childbirth is a deeply rooted practice influenced by cultural beliefs, accessibility, and perceived efficacy in many parts of the world [1]. Traditional medicines refer to natural remedies used by women for any reason during the process of childbirth to support maternal health and wellbeing [2]. Traditional medicine use, is widespread, with some women using them during pregnancy, labour, and postpartum. This reliance on traditional remedies is particularly common in developing countries, where access to formal healthcare may be limited. Globally, maternal mortality remains a major public health concern, with an estimated 322,000 deaths annually, 99 per cent of which occur in developing nations [3,4]. The leading causes include postpartum haemorrhage, eclampsia, obstructed labour, and sepsis. Some of these complications may be exacerbated by the use of traditional medicine [5]. While motherhood is often a positive and fulfilling experience, for some women, it is associated with suffering, ill-health, and even death [6]. In Zambia, many communities believe that the use of traditional medicine can alleviate these challenges and contribute to a more positive experience of motherhood. Some communities believe that use of traditional medicine can assist in making motherhood a good experience for all women [7,8]. Traditional medicine has been practiced in Zambian communities for centuries, predating the advent of modern obstetric interventions. Cultural beliefs have a lot of influence in pregnancy and childbirth practices, and shape decisions regarding traditional medicine use. These cultural practices and beliefs, play a significant role in shaping the maternal health experiences of women in Zambia

Women take different types of traditional medicines for various reasons and people from different cultural backgrounds use traditional medicine as a means to augment, initiate and hasten labour [9–11].

Women prefer traditional medicine because of its intrinsic qualities, uniqueness, holistic use, accessibility, affordability and easy administration. Traditional medicine continues to be an alternative care available, particularly for women in rural areas of developing countries [12].

Traditional birth attendants sometimes rely on the use of certain herbs for their beneficial effects, such as to tone the uterus muscle, induce labour, remove a retained placenta and manage postpartum bleeding. However, the safety of herbal drugs is particularly important in pregnant women [13].

In Zambia, the traditional healthcare system that offers help to pregnant women includes: herbalists, traditional birth attendants (TBAs), spiritual healers, magicians and elderly women in the communities and families. The main source of information on herbal use has familial inclinations (mother, grandmother and aunties), traditional

healers, friends or neighbours and social media. This information is often passed on from one generation to another through verbal communication. The woman's maternal family has the mandate to prepare their daughter for delivery hence the responsibility of looking for the herbs and performing the rituals [10,14].

Meanwhile, there remains a significant gap in comprehensive scientific research exploring the experiences and perceptions of women regarding the use of traditional medicines during pregnancy. This highlights the need for further investigation into how these practices are viewed and experienced by women themselves. The reliance on informal sources such as familial knowledge, traditional healers, and social media, for information about herbal use in pregnancy, coupled with the lack of formal documentation, raises questions about the potential risks and benefits of these practices. Furthermore, little is known about how women perceive the use of traditional medicine alongside modern maternal healthcare services or how these experiences shape their healthcare choices during pregnancy and childbirth. This paper highlighted the need for further research to better understand women's experiences and perceptions of traditional medicine in maternal care, particularly how they navigate its use alongside modern medical practices in Zambia.

## Methods

### Study design

This study was an explanatory qualitative study that explored women's experiences and perceptions towards the use of traditional medicine in selected areas of Zambia. All the women that used traditional medicine during the process of childbirth qualified as participants for this study. The study was conducted in Lusaka, Solwezi and Kaoma districts.

### Study Setting

The study was carried out in three provinces of Zambia that were identified as having the highest maternal mortality burden: Lusaka, North-Western, and Western provinces. Lusaka district is the provincial headquarters of Lusaka Province and a central hub for healthcare services, the research was done at Chawama and Kanyama level one hospitals. The research in Solwezi District, in the North-Western Province, was conducted at two facilities, namely; Solwezi Urban and Kazomba Health Facilities. These sites were selected for their geographic diversity, with Solwezi Urban representing an urban setting and Kazomba serving a more rural community. Both facilities offer maternal and child health services, making them ideal locations for this study. The final site was in Kaoma District in Western Province. The research was conducted at Mangango Hospital Affiliated Health Centre. The facility was selected due to its strategic importance in providing maternal and child health services. It serves as a key referral centre for a large population from surrounding rural areas. This selection process provided a diverse representation of healthcare access, allowing for a more comprehensive understanding of the healthcare needs and challenges in both rural and urban settings.

### Study Participant

This study had two types of participants, 192 women who had used traditional medicine during pregnancy and childbirth and 96 women who were suppliers of traditional medicine in their communities. Eligibility criteria for women who used traditional medicine included being of reproductive age (18–49 years), and having used traditional medicine during pregnancy or childbirth. For traditional medicine suppliers, eligibility required active involvement in preparing or providing traditional remedies for maternal health.

### Sample size determination and sampling strategy

The sample size was determined based on thematic saturation, meaning data collection continued until no new themes emerged from participant responses. The qualitative sample was drawn from a larger quantitative study, ensuring a diverse representation of women with first-hand experiences using or supplying traditional medicine.

PLOS Global Public Health

A purposive sampling strategy was used to ensure the inclusion of women with relevant experiences. Women were selected based on their knowledge and engagement with traditional medicine in maternal health. Snowball sampling was also employed, particularly for identifying traditional medicine suppliers, as some were not easily accessible through formal health structures.

### Data collection

A Focus Group Discussion (FGD) guide and an In-Depth Interview (IDI) guide was used to collect data between 2020 and 2021. This study conducted focus group discussions with women who had used traditional medicine. In-depth interviews (IDIs) were used to collect data from elderly women who supplied traditional medicine. The discussions explored key themes related to their experiences and perceptions of traditional medicine in maternal health. Sixteen (16) focus group discussions were conducted with women who used herbal medicines during the process of child birth (n = 192) and 8 with older women that supplied traditional medicine at community level (n = 96).

### Data analysis

A thematic approach was used to analyse the qualitative data. The in-depth interviews (IDIs) and focus group discussions (FGDs) were recorded with participant consent. Audio recordings were transcribed verbatim and translated into English, as the original interviews and discussions were not conducted in English. Data were coded and organised using NVivo 10 (QSR international). Thematic analysis is a flexible approach which enabled the researcher to identify, analyse and report patterns within the data [Dawadi, 2020]. After closely reading the transcripts and observation notes, the researcher, with the guidance of the supervisors, developed a codebook capturing deductive (i.e., from topics probed in the semi-structured guides) and inductive (i.e., emerging from participant narratives) themes. Codes were then grouped to form themes. Thematic patterns identified summarised into key findings presented herein.

### Ethical consideration

Ethics clearance was obtained from the University of Zambia Biomedical Research Ethics Committee (UNZABREC) Reference number **632–2019.** Written permission to conduct the study was obtained from the respective District Health Offices (DHO). The purpose and nature of the study were explained to all participants, emphasising that participation was voluntary, and they could withdraw at any time without any consequences or loss of privileges.

Those who agreed to take part in the study provided informed consent by signing a consent form. Women who could not read were asked to provide thumbprint consent after the form was read aloud to them. To ensure privacy and confidentiality, all interviews and discussions were conducted in a private setting, and participants' identities were anonymised in all records and reports.

## Results

Results were drawn from a total of 288 participants, including 192 women who had used traditional medicine during childbirth and 96 community members who supplied these remedies as seen in **Table 1** below.

To provide a deeper understanding of this practice, the findings have been organised into key themes that reflect women's experiences and perceptions regarding the use of traditional medicine during childbirth (see **Table 2**).

### Women's experiences with traditional medicine in childbirth

#### 1. Methods of Administration

Women described various ways of administering traditional medicine, each deeply rooted in cultural beliefs and practices. The most common methods included oral ingestion, vaginal insertion, and medicinal tattooing. These methods were used to ensure a smooth labour process, reduce pain, and prevent complications.

**Table 1. Summary of Characteristics of Participants.**

| Province | District | Participant Group | Sample Size (n) |
|---|---|---|---|
| **Lusaka** | Lusaka | Women Who Used (WWU) traditional medicine | 96 |
| | | Women Who Supply (WWS) traditional medicine | 48 |
| **North Western** | Solwezi | Women Who Used (WWU) traditional medicine | 48 |
| | | Women Who Supply (WWS) traditional medicine | 24 |
| **Western** | Kaoma | Women Who Used (WWU) traditional medicine | 32 |
| | | Women Who Supply (WWS) traditional medicine | 24 |
| **Total** | – | **All Participants** | **288** |

**Table 2. Summary Table of Themes and Subthemes.**

| Theme | Sub-theme |
|---|---|
| **1.** Methods of Administration | a) Oral Ingestion<br>b) Vaginal Insertion<br>c) Medicinal Tattooing and sitz baths |
| 2. Timing of Use | a) During Pregnancy<br>b) During labour<br>c) Postpartum |
| 3. Experiences | a) Positive Experiences<br>b) Negative Experiences |
| 4. Perceptions of Traditional Medicine | a) Perceived Benefits<br>b) Avoiding C-Sections |
| 5. Risks and Barriers | a) Fear of Hospital Staff and Stigma<br>b) Influence of Family and Social Expectations |
| 6. Gendered Expectations | a) Postpartum Healing<br>b) Fertility Expectations and cosmetic purposes |
| 7. Cultural Beliefs | a) Infidelity and Pregnancy Complications<br>b) Shame and Reputation |

### a) Oral Ingestion: Drinking Herbal Mixtures for Labour Induction and Pain Relief

For many women, drinking herbal mixtures was seen as an essential step in preparing for labour. Herbs were either mixed with water for the maternal woman to drink or the herbs were added to porridge for the woman to eat. Women were usually given traditional medicine before leaving home to go and deliver at the hospital. The administration of these remedies followed strict cultural protocols that reinforced their perceived effectiveness.

*"Some of us delivered at home and when you are home, they will give you herbs mixed with water for you to a drink or they mix the herbs with porridge and ask you to eat, they will also give you some medicine to apply on the tummy and add to bathing water."*- (FGD, WWU 7)

The belief in the power of herbal medicine to accelerate labour was so strong that women were given these remedies at the onset of contractions. Women were sometimes advised to perform some rituals and traditional customs while drinking herbs. The account illustrates the ritualistic nature of traditional medicine use. The act of dropping the cup on the belly and avoiding looking back, symbolises a spiritual transition into motherhood and a belief that following these customs ensures a successful and complication-free delivery.

*"As the pain increased, they said we could go to the hospital. As we were leaving the house, I was given a very bitter tea which had herbs. I was helped to drink this herb from the door as we exited the house. I was told not to touch the cup, but my sister did it for me and after I finished, I was told to drop the cup on my belly and never to look backward up to the hospital."* – (FGD, WWU 5)

Some women described a structured, phased approach to taking herbs, beginning months before labour to allow the body to get accustomed to herbs and prepare for birth. The quote shows a strong trust in herbal medicine as a preparation tool for labour. The emphasis on timing and consistency in consuming these remedies reflects a deeply ingrained belief in their effectiveness, even in the absence of scientific validation. Traditionally there was a strong belief that the bitter the medicine the better. Bitterness is a measure for effectiveness of the herbal medicine.

*"…for me, I used "mono" the time I was pregnant. I soaked the roots when I was eight months and I even started drinking. I took that for two weeks and I stopped then again, I drunk some during the nineth month. Then the day labour contractions began, I made porridge and put "mono" and drunk it from the door. I even left the cup and someone else came to pick it up."* – (IDI, WWU 2)

**b) Vaginal Insertion and Sitz Baths: Perceived Benefits for Labour and Postpartum Healing**

Women also reported using vaginally inserted herbal remedies, particularly to lubricate the birth canal during labour and restore vaginal tightness during postpartum. These practices were guided by the belief that childbirth causes irreversible changes to a woman's body, which must be corrected using traditional medicine. The scenario highlights gendered expectations regarding women's bodies. The pressure to regain vaginal tightness stems from sociocultural norms that associate a "tight" vagina with fidelity, attractiveness, and marital stability.

*"My grandmother told me that during labour, my vagina will stretch so. I was given some medicine to insert in my vagina soon after delivery to tighten my vagina."* - (IDI, WWS 3)

After delivery a warm, shallow bath for the perineal area is offered to the maternal woman for pain relief, to reduce inflammation, tighten vaginal wall and help keep the perineal area clean after childbirth. A sitz bath is a type of treatment that is done by sitting the maternal woman in ice cold water which can be mixed with salt or traditional medicine and clean fresh water.

*"I tell you seating in a dish full of fresh water was very soothing and it really helped reduce on the pain. Though it was interchangeably done with salty water which was painful and promoted fast healing"* (FGD, WWU 5)

*"We soak herbs in water once we observe that the pregnancy has reached term and after delivery we ask the woman to be sitting in the water soaking her perineal area completely"* (IDI, WWS 2)

**c) Medicinal Tattooing: A Culturally Embedded Practice for Faster Labour**

In some communities, medicinal tattoos were used as an alternative to oral ingestion or vaginal insertion. These tattoos involved embedding herbal extracts directly into the skin incisions, believed to work faster than drinking or inserting herbs.

*"The way these herbs are sometimes, they will work really fast when they do tattoos, the medicine goes directly in the blood."* – (FGD, WWU 7)

## 2. Timing of Use

Traditional medicine was used at various stages of pregnancy, labour, and postpartum recovery, each driven by cultural beliefs, fears, and perceived health benefits.

### a) During Pregnancy: A Preventive Measure Against Complications

Many women began taking herbs early in the pregnancy stage, and believed that the herbs protected the mother and baby against the risk of complications. This practice reflects a deeply rooted fear of maternal mortality, reinforcing the idea that traditional medicine is essential for survival.

> "*Pregnancy should not be taken for granted, women die, so we start giving different herbs when the pregnancy is small up until delivery.*" – (IDI, WWS 1)

### b) During Labour: Ensuring a Quick and "Normal" Delivery

For many women, herbal medicine was only introduced when labour began, often under pressure from family members. The account below illustrates how the use of traditional medicine is not always a personal choice but a response to social pressure. The fear of shame and judgment compels women to comply, even when they have doubts. The emphasis on a "normal" vaginal delivery reflects wider societal expectations that equate caesarean-sections with weakness.

> "*They waited for me to be in severe pain; that's when my mother and mother-in-law gave me the herbal medicine to drink. I hesitated, but my mother warned me that if anything happened to the baby, I will be a laughingstock as well as the whole family will be put to shame. My mother-in-law kept saying, a woman enough should deliver normally and that the baby should be safe as well and herbs remain helpful if this is to be achieved.*" – (FGD, WWU 2)

### c) Postpartum Period: Smooth Recovery

Herbs were given to the women postnatally to aid recovery as most women explained that after labour they experienced so much pain from caesarean section, episiotomy, instrumental deliveries, and from the breasts in the early stages of breast feeding. Traditional postpartum care was mainly to address pain and promote recovery and also gave a woman a sense of belonging. Incorporating herbal medicine into postpartum care helped women to experience improved physical and emotional well-being, supporting a smoother recovery and transition into motherhood.

> "*Our women go through a lot of pain after delivery, we give them the traditional medicine to help them recover. We don't even allow them to do any work, they have to rest so that they recover.*" – (FGD, WWS 8)

## 3. Experiences of Positive and Negative Outcomes

Women's experiences with traditional medicine varied widely, with some reporting benefits and others experiencing complications. Such positive testimonials create a powerful reinforcement cycle, leading more women to trust and continue using traditional medicine.

### a) Positive Experiences: Trust in Herbal Medicine's Effectiveness

Many women who had successful, fast labours attributed their experiences to herbal medicine, reinforcing its perceived effectiveness.

*"I was in pain at the hospital and my mother came with her friend. They brought me a bitter tea; 15 minutes after I took this tea, all the pain was gone. I progressed so fast. I never laboured. I was given the medicine and in no time, the baby was born. Really, mum and her friend were angels, heaven sent. Unfortunately, that friend of mum died and she was the one who knew the traditional medicine tree. Well all I can say is that the herbs are good, they are natural and has got no side effects kkkkk I am speaking because I have experience. The first pregnancy I never used mmmm i suffered…. mmmm I will always use and I can't even allow my daughter to go in labour without the herbs "* – (FGD, WWU 8)

### b) Negative Experiences: Unregulated Use and Harmful Effects

Despite widespread belief in the benefits of herbal medicine, some women suffered severe complications, including excessive bleeding and infant loss.

*"I was given the medicine by one of my aunties. I experienced a lot of vaginal bleeding and I ended up losing the first baby. Mmmmmm, I don't want to use these herbs again, that woman only God knows. At the hospital, they said it was the herbs that killed the baby. I have three children and I always go to the hospital. I don't use herbs."* - (FGD, WWU 1)

While women trust these remedies, there is no standardisation or medical oversight, increasing the risk of overdose or negative interactions. Despite these risks, social pressure often prevents women from openly questioning the safety of herbal remedies.

### 4. Women's Perceptions of Traditional Medicine in Childbirth

### Perceived Benefits

### a) Faster Labour and Pain Reduction

Many women believed that herbal medicine shortened labour and reduced pain, making it preferable to biomedical interventions.

*"Like for me, I did not take long. I went to give birth from Lusaka's University Teaching Hospital (UTH) because the baby was in breach position. I was at Chawama Clinic, then I was taken to UTH and they said that I was going to be operated on. When I reached UTH, I did not even take long to give birth."*- (FGD, WWU 7)

*"The difference is that for us we prevent the pain being extreme but at the hospital, they give when things get to the worst."* - (IDI, WWS 1)

The statements show a clear preference for proactive pain management rather than waiting for medical intervention.

### b) Avoiding Caesarean-Sections: A Social and Cultural Priority

Many women feared Caesarean sections (C-sections), believing they were dangerous, unnecessary, and associated with marital problems. This fear reflects deeper gender norms where a woman's ability to deliver vaginally is linked to strength, fertility, and social acceptance.

*"The conventional medicine is there but it does not work usually. You can bear me witness. A Woman can go around 7 hours to give birth and they spend the whole day there. So, sometimes, women refuse to be operated on and prefer herbal medicines. So, usually because of the operations done, most women end up using the herbal medicines for*

*steaming, applying in the thighs, drinking, making porridge, applying on the stomach such that once you go to the clinic, it is one time no struggle at all.*" - (IDI, WWS 2)

## 5. Perceived Risks and Barriers

### a) Fear of Hospital Staff and Stigma

Women reported avoiding full disclosure about the use of herbal medicine due to harsh treatment from healthcare workers.

"*The nurses talk about it but the way they ask scares us to open up. You will have taken something from home and the nurses will ask you what you have taken in a very rude and harsh way…They will say. "That is why you die, why did you take this and that?" and then you are feeling so much pain and someone is shouting at you. So, we just say we did not take anything and yet you had taken.*" - (FGD, WWU1)

"*I had experience with two nurses the first one was younger but had a temper. But an elderly person comes and says don't worry if your relatives gave you something, you will be okay. Then those who are rude when in pain will say, you see it is the things that you took that caused you this. It will deal with you. We are suffering as we go for labour…We have so much fear.*" - (FGD, WWU 8)

"*Mmmmmm, me I was given these herbs at home so when I reached, I found that the nurse was busy shouting at someone, even you, are you sure you can admit it if you are asked. Mmmmmm you just say no and now you are even more scared because you are now in a dilemma.*" – (FGD WWU 1)

Such experiences highlight a gap in patient-provider communication. Fear of judgment and mistreatment prevents women from discussing their herbal medicine use, increasing risks of drug interactions and complications.

### b) Influence of Family and Social Expectations

Social norms strongly dictated the use of herbal medicine. Women felt pressure from family members to conform, fearing shame or blame if they did not.

"*So, a woman is not supposed to spend a lot of time in labour as the pain can kill someone if it prolongs. That is a very difficult time so if they tell as the woman has been sent to the hospital because the clinic can't handle, we also get worried and we want to be part of helping the woman, so as they prepare the woman, we also go in the bush to looks for the herbs and the first tree we meet whose root crosses the path is the herb that will help the mother. We need to cut the root crossing the path and we boil it and make a herbal drink. Then we rush to the hospital this is done very fast this is to make sure that we don't find our woman in theatre. The herbs have saved so many women from caesarean section. Caesarean section is destroying our women and their marriages it even gives them a reduced lifespan*". - (IDI, WWS 6)

"*My mother said if I didn't take the herbs, I would bring shame to the family.*" - (FGD, WWU 3)

Women who supply traditional medicine had reservations on supplying traditional medicines to women who were impregnated by married men. Besides, women who had a history of being impregnated by married men or had unwanted pregnancies or pregnancies out of wedlock lamented that it was very difficult to get traditional medicine from relatives and community elders. Social norms and stigma play an important role in reproductive health behaviour and decision-making for pregnant women. One woman highlighted how her friends helped her secure traditional medicine because the family did not accept her pregnancy.

*"The family was very annoyed that I was pregnant,my family never discussed any thing concerning traditional medicine, at church I was excommunicated and only my friends would give me hints on how to go about pregnancy issues, my friends could also helped me source for some herbal medicines to protect me during labour, I remember a friend I met during antenatal introduced me to a woman I used to buy herbs from"- (FGD, WWU 9)*

**Gendered expectations**

**a) Postpartum Healing: Restoring Fertility and Female "Readiness"**

Herbs were also commonly used after childbirth to help the mother recover, restore fertility, and tighten the vagina. Such beliefs reinforce gendered expectations and control over women's reproductive health, making it difficult for them to refuse herbal treatments.

*"We give herbs to our women even post-delivery, the women's body has to return to at least its normal position. We give women to also return fertility so they are prepared for the next pregnancy." – (IDI, WWS 5)*

This highlights deeply ingrained gender norms where a woman's body is expected to return to a certain state after childbirth, not only for her health but for marital and reproductive expectations.

**b) Fertility Expectations and Cosmetic purposes**

Lactational amenorrhea is viewed as infertility in the traditional circles therefore, post-delivery the women are given traditional medicine to help her retain fertility through the show of menstruation. Additionally, women are given herbs for cosmetic purposes to help restore their beauty so that they continue to be attractive to their partners.

*"The pregnant woman should start applying the oil in the 6th month of pregnancy so that they prevent stretch marks kkkkkkkkkk the woman should maintain their beauty, men can leave them"(FGD, WWS, 7)*

**Cultural beliefs**

**a) Infidelity and Pregnancy Complications.**

Some other sentiments were based on beliefs of infidelity and how it affects labour and delivery to the point of death if either the woman or her husband had sexual relations with other people and did not use traditional herbs to prevent death during labour or delivery.

*"We believe in our culture and a belief across is that men sleep around; they fail to manage one woman. So, when they sleep around, they give us 'inchila'(complicated labour). We are the ones that suffer the most and for us to survive, we need to take these herbs. Ichalo chalishupa (Life is hard) she smiles and say emmmmm its protection." - (FGD, WWU 2).*

*"On that one, sometimes it is because of their husbands having sexual relations with other girls when there woman is pregnant. Secondly, the vagina does not expand enough for the baby to come out, but the Doctors and Nurses try to tell women to push, but if that woman has no strength to push, they end up dying." - (IDI, WWS 6)*

On the other hand, the supplier said the man is given medicine for cleansing if he opens up and says he indulged himself in sexual activities.

*"If a man cheats while his wife is pregnant, the woman might struggle in labour. We give herbs to either the man or woman to protect her."* - (IDI, WWS 6)

*"We give the man traditional medicine and we ask the man to march naked going around his house seven times at midnight."* - (IDI, WWS 1)

**b) Shame and reputation**

Most Families with women that experience complicated pregnancies do not understand the nature of childbearing, as a result they blame the maternal woman for any complication they face. On the other hand, the Women themselves who experience pregnancy complications may have self-stigma. Women and their families may feel guilty or blame themselves, questioning whether they could have done something differently. To save themselves from such shameful experiences pregnant women accept to be given herbal medicines to prevent any pregnancy related complications.

*"We give the woman traditional medicine so that we avoid putting the family into shame, once a woman dies from pregnancy complications or experiences certain complications. Society will place the blame on the family and on us the elderly women in the community."* - (IDI, WWS 1)

The findings further highlighted complex interplay between cultural traditions, perceived health benefits, and biomedical risks. These insights underscore the need for culturally sensitive healthcare approaches that bridge the gap between traditional and biomedical maternal care to ensure safer childbirth experiences for women.

## Discussion

This study showed that women have various experiences towards the use of traditional medicine in selected areas of Zambia.

This study revealed that information about herbs used during childbirth was mainly spread by the family and elderly trusted women in the community. It was the responsibility of the family to transfer this information from one generation to the other.

Contrary to this finding, some researchers [15–17] have reported different perspectives. Information on herbs of foreign origin is becoming increasingly accessible through extensive marketing processes such as advertisements on various media platforms, including Facebook, other social media sites, television, and radio. These results vary with the current study's results due to differences with the study settings. Thus, while some studies frame herbal use as initiated by "the woman's own initiative" or recommendations from friends, our findings further reveal that for many, compliance is socially enforced, not freely chosen.

Interestingly, this study revealed that friends were helpful and supplied the herbs in instances that the pregnancy was illegal and was disapproved by the family. This finding is in line with the findings of Baladi et al., [18] that the use of herbal medicines was most commonly initiated after recommendations from friends in instances where it was an adolescent mother or school going mother. Further, scholars such as Nyaloko et al., [19] highlight that herbal use during childbirth was linked to relationships and trust issues. This was observed in the current study as many women explained that their main reason for trusting such advice was that they knew the supplier very well.

Various researchers [20,21] concluded that the use of herbal medicine during pregnancy and childbirth in rural areas is heavily influenced by women's previous labour experiences, fears of abandonment during childbirth, and the information they receive from their social networks. This study's findings suggest that community-led pathways, such as leveraging the influence of peers and traditional birth attendants, can be effective in shaping women's decisions regarding the use of herbal medicine.

The Results detail specific modalities of herbal administration, oral ingestion, vaginal insertion, medicinal tattooing, and sitz baths, not merely as delivery mechanisms, but as culturally codified rituals imbued with symbolic and functional significance. For instance, the act of dropping the cup on the abdomen and refusing to look backward upon leaving for the hospital to deliver is not a peripheral custom; it constitutes a liminal rite of passage, marking the woman's transition into motherhood under the perceived protection of ancestral or spiritual forces. The vaginal insertion of herbs and the use of sitz baths to restore postpartum vaginal tightness reflect not only therapeutic intent but also deeply gendered social expectations. These practices are driven by the belief that a woman's social and marital value is contingent upon the physical state of her body, a finding that transcends the notion of "personal philosophies and attitudes" to reveal how traditional medicine functions as a tool of social conformity [20,21]. It was observed through this study that traditional medicines are different and are used differently depending on the drugs, the condition of the woman and the place of delivery. Most of these remedies were taken in the third trimester, at the beginning, mid or at the end of the last trimester or when labour had begun. Most of these drugs were taken orally, others were tattooed, others through squats and sitz baths, some women could apply topically and others would apply in the birth canal. Notably, most of these drugs were mixed with porridge or women would make either a hot or cold drink. These observations were very common as explained by different authors [19,22–25]. Although the women in the current study felt the processes of taking these were safe, multiple authors [19,22–25] had issues with the safety of the processes used. This variability could be linked to the difference in the study designs. Given these concerns, further research is needed to scientifically assess the safety and efficacy of different traditional medicine preparation and administration methods. Public health interventions should focus on educating women on safe maternal health practices.

The belief that herbal medicines have no side effects, reduce pain, lead to a quick, smooth and safe delivery process, emotional and family stability, social acceptance, and positive childbearing experiences were the most commonly cited reasons for the use of herbal medicine in this study. The argument by pregnant women using herbs was that medicine from the hospital is not very effective in dealing with most pregnancy-related complications, no wonder they will rush for surgical interventions such as caesarean birth. Most pregnant women in our study and other studies elsewhere [20,21] often explained the naturality of herbs to mean safety and did not envisage any side effects consumed during pregnancy. The perceived effectiveness of herbs is more important to pregnant women than the possible side effects. Women further highlighted that herbs and traditional practices were commonly used to prevent complications related to pregnancy and childbirth, which was not a common feature for hospital interventions. Various authors [19,22–25] pointed out that there are conflicting and insufficient evidence regarding the efficacy and safety of traditional and herbal remedies for labour induction stating that safety, is an important feature of any treatment and intervention by many pregnant women. The perceived safety and effectiveness associated with herbs is contested since there is no substantial empirical evidence and verifiable data explaining the safety of most herbs used by pregnant women, especially those in rural areas [23].

This study highlighted that women used herbal medicine during pregnancy, labour and postnatally because of different pregnancy related problems. Hence, the pattern of herbal medicine use throughout the continuum of pregnancy noted in this study is consistent with findings from Zambia's neighbouring country Malawi done by [12], Who's study revealed that herbal medicine use was observed throughout the continuum of pregnancy, but interestingly this study notes higher use during labour for women in rural areas. However, it is difficult to ascertain whether the difference in prevalence are caused by variability in study design, setting, data collection, and sampling techniques or whether they represent true differences in herbal medicine use.

The current study's findings revealed a complex interplay of perceived benefits, spiritual beliefs, and social pressures influencing women's decisions to use traditional medicines. Many participants reported using herbs to "ease labor," "open the birth canal," or "prevent complications," reflecting longstanding cultural knowledge [10,26].

The pressure to avoid Caesarean sections described as "destroying our women and their marriages" further illustrates how biomedical interventions are not evaluated solely on clinical grounds, but through the lens of cultural stigma and

gendered identity. This fear is not abstract; it is coercive, as women report being warned that refusal to take herbs will "bring shame to the family". According to this study, women would get herbal drugs even from strangers, especially in desperate and crisis situations such as referral, prolonged labour and any complication or condition that would prevent a caesarean section from happening. This finding is not in tandem with other researchers [20,21], who stated that factors influencing the use of herbal medicines were related to both facilitating conditions such as satisfaction, trust and norms such as personal philosophies and attitudes. The difference could have been attributed to the difference in the study settings as the Malawi study only concentrated on the rural area were as the current study was done in a rural and urban setting. Given these findings, public health initiatives should focus on educating women on safe maternal health practices. Additionally, integrating culturally sensitive maternal health education within communities may help address fears related to medical interventions like caesarean sections.

Additionally, this study notes the contribution of the formal healthcare system in actively perpetuating secrecy and amplifying risks among pregnant women. Women across multiple focus groups consistently recounted experiences of being "shouted at" by nurses, blamed for their pain, and interrogated in "a very rude and harsh way" when suspected of using herbal remedies. This institutionalized hostility fosters a climate of fear that actively discourages disclosure a critical barrier to safe clinical management. As one participant poignantly noted: "So, we just say we did not take anything and yet we had taken." This is not merely a breakdown in communication; it reflects a systemic failure wherein healthcare facilities, intended as sanctuaries of care, are instead experienced as sites of judgment and reprimand. While Ampomah et al. [20] and Adeoye and Etuk [21] frame trust and social norms as key determinants of health-seeking behaviour, the data revealed that mistrust is not solely culturally embedded it is iatrogenic, cultivated by the very structures and actors entrusted with maternal safety. Consequently, public health interventions must extend beyond educating women, to include mandatory training for healthcare providers in culturally competent, trauma-informed communication, transforming institutional cultures to prioritize psychological safety alongside clinical safety. Additionally, collaboration between traditional practitioners and healthcare professionals could help establish safer practices and integrate beneficial remedies into maternal care under proper supervision.

Cultural beliefs represent the society communality and have a lot of influence regarding traditional medicine use during the process of childbirth. Each cultural grouping has certain beliefs that hold them together and they have various factors that lead to herbal medicine use [9–11]. This study highlighted factors such as pain, age, suspected infidelity, previous experience, and distance to facilities as key influences on herbal use. These findings resonate with [1] in Malawi and with Ampomah et al. [20] and Adeoye and Etuk [21] on the role of prior experience and social networks. However, the data collected in this study add critical nuance, the belief that herbs prevent death caused by a husband's infidelity reveals how traditional medicine is also used to manage perceived spiritual and moral threats, not just biomedical ones. The ritual cleansing of men who confess infidelity, requiring them to "march naked around the house seven times at midnight" further illustrates the gendered asymmetry in how reproductive risk is managed and assigned.

On the other hand a qualitative descriptive study that investigated the dynamics surrounding the use of herbal medicines by women during pregnancy and childbirth in rural Malawi revealed that factors influencing the use of herbal medicines included, some characteristics such as age, ethnicity, gravida, maternal occupation, family income and maternal education level, [1]. In both studies the common finding is age and the difference in the other study's findings could be because the studies were done in different setting dynamics. From the different sources of literature reviewed it is evident that traditional medicine use is a common practice in most communities. Hence the use of traditional medicine during pregnancy is a complex issue that requires careful consideration of both potential benefits and risks. This calls for all stakeholders involved in maternal health to invest in women education and community education on the use of traditional medicines during pregnancy. The community needs sensitization on significant risk factor for maternal mortality that may potentially cause serious complications such as sepsis, uterine rupture and postpartum haemorrhage. These risks are compounded by a lack of safety data, unregulated dosage, and potential contamination of herbal remedies.

## Strengths and limitations

One of the strengths of this study is that it provides first-hand insights into women's lived experiences and perceptions regarding the use of herbal medicine during pregnancy and childbirth. By capturing both users and suppliers, this study presents a comprehensive perspective on the cultural and social factors influencing traditional medicine use. Additionally, conducting the study in diverse settings (both rural and urban) enhances the generalisability of the findings within the Zambian context.

However, the study has some limitations. First, self-reported data may be subject to recall bias, as participants might not accurately remember details about their herbal medicine use. Secondly, there was no clinical assessment of the safety and efficacy of the herbs mentioned, meaning findings rely solely on participants' perceptions rather than biomedical validation. Thirdly, the study's qualitative approach limits broader generalisability beyond the study population. Future research should include scientific analysis of commonly used herbal medicines to determine their pharmacological properties, potential risks, and benefits.

## Conclusion

The use of herbal medicine during pregnancy and childbirth is a common phenomenon. Women used one or more herbal medicines during pregnancy. Results of this study showed that herbal medicinal products are used to widen the birth canal, augment labour, speed delivery and reduce pain. Some women used herbal medicine in the first trimester while others used it in the second or third trimester or throughout pregnancy. This study emphasises the importance of understanding local contexts and information sources to inform health interventions and support safe maternal health practices in Zambia.

## Author contributions

**Conceptualization:** Wanga Zulu.

**Data curation:** Wanga Zulu.

**Formal analysis:** Wanga Zulu.

**Investigation:** Wanga Zulu.

**Methodology:** Wanga Zulu.

**Project administration:** Wanga Zulu.

**Resources:** Wanga Zulu.

**Supervision:** Joseph Mumba Zulu, Charles Michelo, Choolwe Jacobs.

**Validation:** Wanga Zulu.

**Visualization:** Wanga Zulu.

**Writing – original draft:** Wanga Zulu.

**Writing – review & editing:** Wanga Zulu, Joseph Mumba Zulu, Charles Michelo, Choolwe Jacobs.

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
