## [Decision Letter · Decision Letter 0]

4 Sep 2025

PGPH-D-25-01444

Women’s Experiences with the Use of Traditional Medicine during Childbirth in Selected Areas of Zambia

Dear Dr. Zulu,

Thank you for submitting your manuscript to PLOS Global Public Health. After careful consideration, we feel that it has merit but does not fully meet PLOS Global Public Health’s publication criteria as it currently stands. Therefore, we invite you to submit a revised version of the manuscript that addresses the points raised during the review process.

The manuscript has been evaluated by three reviewers, and their comments are available below.

We look forward to receiving your revised manuscript.

Kind regards,

Jianhong Zhou

Staff Editor

Journal Requirements:

2. In the online submission form, you indicated that “Due to ethical and privacy concerns, the full qualitative data set (transcripts from focus group discussions) cannot be made publicly available. However, de-identified excerpts relevant to the study findings are available upon reasonable request. Interested researchers may contact the corresponding author, Wanga Zulu (zulunkuwa@gmail.com), to request access to the data.”

3. Uploaded as supplementary information.

3. Your manuscript is missing the following sections: Discussion. Please ensure these are present, and in the correct order, and that any references to subheadings in your main text are correct. An outline of the required sections can be consulted in our submission guidelines here: 

https://journals.plos.org/globalpublichealth/s/submission-guidelines#loc-parts-of-a-submission

Reviewers' comments:

Reviewer's Responses to Questions

**Comments to the Author**

1. Does this manuscript meet PLOS Global Public Health’s publication criteria? Is the manuscript technically sound, and do the data support the conclusions? The manuscript must describe methodologically and ethically rigorous research with conclusions that are appropriately drawn based on the data presented.

Reviewer #1: Yes

Reviewer #2: Yes

Reviewer #3: Partly

2. Has the statistical analysis been performed appropriately and rigorously?

Reviewer #1: N/A

Reviewer #2: Yes

Reviewer #3: N/A

3. Have the authors made all data underlying the findings in their manuscript fully available (please refer to the Data Availability Statement at the start of the manuscript PDF file)?

Reviewer #1: Yes

Reviewer #2: Yes

Reviewer #3: No

4. Is the manuscript presented in an intelligible fashion and written in standard English?

Reviewer #1: Yes

Reviewer #2: Yes

Reviewer #3: Yes

5. Review Comments to the Author

Reviewer #1: The manuscript is well written but there are a few typos. Line 392 should be "Discussion". Line 434 should be "sitz baths" not "sizi baths". The important finding on line 402 and 403 that "friends were helpful in instances where the pregnancy was illegal and disapproved of by the family" is not highlighted in the results section.

Reviewer #2: July 2025

The manuscript is well put together and can be summarised as

"Cultural beliefs related to pregnancy, delivery, and child-bearing are passed down from generation to generation through guided and unguided learning. Most researchers emphasise that the safety of traditional medicines is particularly important for pregnant women. However, research on Zambian women’s experiences and perceptions of using traditional medicine remains limited, leaving a gap in understanding its impact and effectiveness on maternal health. To better understand women’s experiences and perceptions of traditional medicine in maternal care, particularly how they navigate its use alongside modern medical practices in Zambia. A a qualitative study that explored women’s experiences and perceptions towards the use of traditional medicine in selected areas of Zambia, emphasizes the importance of understanding local contexts and information sources to inform health interventions and support safe maternal health practices in Zambia."

These are quite significant for the discipline and properly placed in the context of the previous literature. The literature review is robust. The data and analyses are satisfactory and the details of the methodology are sufficient to allow replication.

Keep up the good work.

Reviewer #3: Dear authors,

Thank you for the opportunity to read your interesting manuscript. The study seems well planned but I have some comments to the reporting.

Line 33: what do you mean by “rolling”?

Line 37: Is “excessive bleeding” mentioned in the Results-section of the paper? (if so, everything is fine, and I apologise for the comment).

Line 60: 2014 is a long time ago – is it possible to find new data?

Lines 71-76: You talk about “Traditional medicine” and “Alternative therapies” – what is the difference?

Line 77: What do you mean by “abuse” here?

In general, about the introduction: The references are relevant and good, but it is more like a series of quotes than a flowing text. I suggest that you work a bit more with this part. Think more like “you are telling a story to make me read your paper – not just giving me some quotes”.

Line 137: what do you mean by “rolling”?

Line 163: “translated and transcribed” – below you state it the other way round. What is correct and where does it fit best in?

Line 165: The “in-depth interviews” are not mentioned under “Data collection” – did you do any? With whom? Describe under “Data collection” if you did.

In general, about the results: why are the women suddenly “girls” for the quotes? Where is the abbreviation WHS defined? Abbreviations should always be defined on first use. Some of the quotes are a bit difficult to understand – did you try to translate “the way people speak” or should you work a bit more on the translations?

Line 206: “Herbs were either put or drunk in tea…” put where? What do you mean? “… or taken just before leaving” – but taken how?

Line 246: “…. or inserting herbs” Inserting where?

Line 293: What do you mean by “…her friend is late the one who knew the tree”?

Line 309: UTH?

Line 314: Why suddenly “Group 1”? is that pregnant women or suppliers of treatment?

Line 345: What is “Ba WZ”?

Lines 377-8: Ba WZ again?

Line 386: “…match..” – did you mean “…march…”?

In general, about the discussion: you should not discuss any results not given in the “Results” section! The discussion needs to be rewritten to mirror the results. Where in the results do you find anything about “correctly classifying, describing and identifying herbal products (ref lines 396-7)? All the numbers given in lines 410-6 are not in the Results so can not be discussed here and numbers or percentages are not the point in a qualitative paper. You repeat yourself a lot in the beginning of the discussion talking about “family and friends” several times.

Line 413: where is the abbreviation TP defined?

Line 434: squats and “sizibaths” are not mentioned in the results, so should not be discussed here.

Line 444: It is a good idea to “educate women about potential risks” but what do we KNOW about potential risks?

Lines 448-460: You need to comment on HOW the other studies are similar to or different from yours – not just mention them.

Lines 461-6 and 467-71: are the results you discuss here actually mentioned in the Results-section?

Lines 481-6: This is the really important part to discuss.

Line 501: what do you want to study in a clinical trial?

6. PLOS authors have the option to publish the peer review history of their article (what does this mean?). If published, this will include your full peer review and any attached files.

**Do you want your identity to be public for this peer review?** For information about this choice, including consent withdrawal, please see our Privacy Policy.

Reviewer #1: No

Reviewer #2: No

Reviewer #3: No

---

## [Decision Letter · Decision Letter 1]

5 Nov 2025

Women’s Experiences with the Use of Traditional Medicine during Childbirth in Selected Areas of Zambia

PGPH-D-25-01444R1

Dear Mrs Zulu,

We are pleased to inform you that your manuscript 'Women’s Experiences with the Use of Traditional Medicine during Childbirth in Selected Areas of Zambia' has been provisionally accepted for publication in PLOS Global Public Health.

Please note: Reviewer 3 has a minor comment regarding one of the tables, which you can update during final formatting checks.

Best regards,

Julia Robinson

Executive Editor

Reviewer Comments (if any, and for reference):

Reviewer's Responses to Questions

**Comments to the Author**

1. If the authors have adequately addressed your comments raised in a previous round of review and you feel that this manuscript is now acceptable for publication, you may indicate that here to bypass the “Comments to the Author” section, enter your conflict of interest statement in the “Confidential to Editor” section, and submit your "Accept" recommendation.

Reviewer #1: All comments have been addressed

Reviewer #2: All comments have been addressed

Reviewer #3: All comments have been addressed

2. Does this manuscript meet PLOS Global Public Health’s publication criteria? Is the manuscript technically sound, and do the data support the conclusions? The manuscript must describe methodologically and ethically rigorous research with conclusions that are appropriately drawn based on the data presented.

Reviewer #1: Yes

Reviewer #2: Yes

Reviewer #3: Yes

3. Has the statistical analysis been performed appropriately and rigorously?

Reviewer #1: N/A

Reviewer #2: Yes

Reviewer #3: N/A

4. Have the authors made all data underlying the findings in their manuscript fully available (please refer to the Data Availability Statement at the start of the manuscript PDF file)?

Reviewer #1: Yes

Reviewer #2: Yes

Reviewer #3: No

5. Is the manuscript presented in an intelligible fashion and written in standard English?

Reviewer #1: Yes

Reviewer #2: Yes

Reviewer #3: Yes

6. Review Comments to the Author

Reviewer #1: (No Response)

Reviewer #2: Satisfactory

Reviewer #3: Table 1: (WWS) is missing for Kaoma.

7. PLOS authors have the option to publish the peer review history of their article (what does this mean?). If published, this will include your full peer review and any attached files.

**Do you want your identity to be public for this peer review?** For information about this choice, including consent withdrawal, please see our Privacy Policy.

Reviewer #1: No

Reviewer #2: No

Reviewer #3: No
